# The Feud over Lactate and Its Role in Brain Energy Metabolism: An Unnecessary Burden on Research and the Scientists Who Practice It

**DOI:** 10.3390/ijms26094429

**Published:** 2025-05-07

**Authors:** Avital Schurr

**Affiliations:** Department of Anesthesiology and Perioperative Medicine, University of Louisville School of Medicine, Louisville, KY 40202, USA; avital.schurr@gmail.com

**Keywords:** astrocyte, bias, brain, energy metabolism, glucose, glycolysis, lactate, neuron, shuttle

## Abstract

Ever since the monocarboxylate, lactate, was shown to be more than a useless end-product of anaerobic glycolysis, the members of the brain energy metabolism research community are divided by two issues: First, could lactate replace glucose as the oxidative mitochondrial energy substrate? Second, should glycolysis continue to be divided into aerobic and anaerobic pathways? This opinion paper examined both the history and the reasons for this division and offered a unifying solution. Some readers may find this paper somewhat slanted, although many aspects of my opinion are backed, whenever possible, by data.

## 1. A Short Introduction

In 1985, George A. Brooks stirred up the exercise physiology field when he published his paper: Lactate: glycolytic end-product and oxidative substrate during sustained exercise in mammals—the “lactate shuttle” [1]. Three years later, my own research group, not being aware of Brooks’ research, published the paper “Lactate-supported synaptic function in the rat hippocampal slice preparation” [2], which stirred up the brain energy metabolism field. Both papers were received with much skepticism by many within the respective research groups, which sparked great debate. Several review articles have been published over the years to summarize the research history of this field and to highlight the controversies [3,4,5,6,7,8,9,10,11,12,13,14,15]. Therefore, this paper only covers the past four decades of research in the field, highlighting the discoveries and the disagreements that continue to divide our community. It lists the possible reasons for these disagreements and offers a path to resolve them. However, being an opinion paper, the reader may sense my own partiality when arguing for or against different points on the two sides of the above-mentioned debate. Nonetheless, whenever possible, arguments are supported by available data.

## 2. The Discoveries That Revolutionized the Study of Skeletal Muscle and Brain Energy Metabolism

Ever since its discovery, lactic acid has earned a bad reputation that prevented scientists from considering it as anything else but a milk spoiler and a waste product of an energy-producing process named anaerobic glycolysis. The glycolytic pathway, the first biochemical pathway to be elucidated in 1940, has remained unchanged ever since. It is still being taught and portrayed as a chain of enzymatic reactions that has two possible outcomes: an aerobic one, composed of 10 reactions, with pyruvate as its end-product, and an anaerobic one, composed of 11 reactions, with lactate as its end-product. Forty-five years after the first ever metabolic pathway was described and split into aerobic and anaerobic outcomes, Brooks [1] published his surprising finding that lactate can be utilized oxidatively by the working muscle and suggested the existence of a shuttle mechanism for that monocarboxylate from the cytosol to the mitochondrion and from one type of muscle fiber to another. The expected headlines that one would anticipate following Brooks’ findings never materialized. His conference paper received only 291 citations, indicating either great skepticism on the part of its readers or the obscurity of the publishing journal. Three years later, and completely unaware of Brooks’ work, our research group published a similar finding about the ability of brain tissue to oxidatively utilize lactate as the sole energy substrate to support neuronal function [2]. Our publication also did not receive much attention (655 citations), although the journal that published the findings is not obscure, even dedicating its cover image to the experimental system used for that discovery. As important as skepticism is in scientific research, ignoring a published breakthrough cannot be considered as such. The common method by which the impact of a given scientific publication is measured is to count the number of times other scientists cite that publication in their own work. Quoting from the University of Illinois Chicago Library web page: “*The process whereby the impact or “quality” of an article is assessed by counting the number of times other authors mention it in their work*”. The impact factor (IF) of journals is also based on the number of citations its published articles received in the publications of scientists. In general, the larger the number of citations a paper garners, the greater the impact. The relatively low number of citations the papers by Brooks [1] and Schurr et al. [2] received, especially over the first eight years following their publication (22 and 63, respectively), indicates that the exercise physiology research community was either oblivious to Brooks’ finding or very skeptical about it, while the brain energy metabolism research community was mainly skeptical about the results of Schurr et al. Interestingly, while most of the research on energy metabolism leading to the proposition of the glycolytic pathway and for three additional decades was performed on muscle tissue, the interest in brain energy metabolism in the past five decades has grown exponentially. This could partially explain why the paper by Schurr et al. received almost three times more citations than the number Brooks’ paper received. Nevertheless, the impact of both papers based on their citation numbers is low, especially during the first eight years after their publication. The lack of enthusiasm about the discovery that lactate is not a waste product, but rather a valuable fuel the cell knows how to utilize, cannot be simply explained by skepticism about its validity. After all, the published data were analyzed and were found to be statistically significant, and the papers were peer-reviewed. As I argued before [11,16], there is another factor that influences the way data such as these are being judged by most of the members of the energy metabolism research community. For decades, that community has worked according to the established Embden–Meyerhof–Parnas’ paradigm, where glycolysis is a metabolic pathway that has two different outcomes, aerobic and anaerobic, ending either with pyruvate or lactate, respectively. Whether one deals with muscular or cerebral energy metabolism today, this paradigm still guides most investigators, teachers, coaches, and consequently, students. Margolis [17] argued that **habit of mind** is the factor that prevents scientists from making the necessary shift from an old paradigm to a new one. However, it seems that this habit of mind drives its owners not only to question the validity of the new data or simply ignore them, but prompts some to take an active position by publishing reasons and arguments that could invalidate those data rather than making an effort to replicate them. This is evidenced by the discoveries of Brooks [1] and Schurr et al. [2], which did not receive much attention for several years. Neither replication of these studies has been performed, nor have skeptic papers been published until more than a decade later. Then, in 1993, Magistretti et al. [18] published a review article where they stated: “*Since lactate can support neuronal activity and synaptic function in vitro, the possibility should be considered that glucose uptake by the brain parenchyma occurs predominantly into astrocytes which subsequently release lactate for the use of neurons*”. The following year, Pellerin and Magistretti [19] published their research on the activation of astrocytic glycolytic utilization of glucose by the excitatory neurotransmitter, glutamate, where the lactate formed thereof is utilized oxidatively by neurons. They have formulated what has become known as the ‘astrocyte–neuron lactate shuttle’ (ANLS) hypothesis. The 1993 and 1994 papers by the Magistretti group cited our paper [2], which brought more attention to the discovery about the ability of lactate to support neuronal function as the sole oxidative substrate of energy metabolism. The Sokoloff group was the first to partially address the ANLS hypothesis [20,21], followed by two more publications [22,23]. All four articles focused on the astrocytic glucose utilization and the distribution of the monosaccharide to neurons without directly addressing the point raised by Pellerin and Magistretti [19] about the astrocytic to neuron transport of lactate and its utilization by the latter. These were the first salvos in what would become a long and concentrated effort to defend the long-held concept that glucose is the obligatory substrate of brain energy metabolism, a concept that neither Brooks [1], Schurr et al. [2], nor Pellerin and Magistretti [19] ever challenged. By contrast, at least two studies, one by Korf [24] and the other by Elekes et al. [25], support the suggested shuttle of lactate between astrocytes and neurons. The ANLS hypothesis has brought much attention to lactate and the role it might play in energy metabolism, as the Pellerin and Magistretti paper [19] has received over 3200 citations by February 2025. Nevertheless, attention did not mean acceptance, evidenced by the many publications over the years that have questioned its validity. By 1998, the first paper questioning the hypothesis was published [26]. Three years later, the viability of the ANLS hypothesis was questioned after a study concluded that the amount of astrocytic lactate produced upon activation could cover only 25% of the energy produced by glucose for neuronal needs [27]. The investigators argued that lactate could not provide the amount of ATP glucose could. Their position, taken by many other researchers, stems from the mantra that glucose is an obligatory energy substrate and there is no other molecule that could replace it. Soon thereafter, Chih et al. [28] published an in vitro study where they concluded that “… *together with observations in the literature that (1) glucose is available to neurons during activation, (2) heightened energy demand rapidly activates glycolysis in neurons, and (3) activation of glycolysis suppresses lactate utilization, suggests that glucose is the primary substrate for neurons during neuronal activation and do not support the astrocyte–neuron lactate shuttle hypothesis*”. This study was followed by an opinion article that essentially debunked the ANLS hypothesis [29]. A study by Shulman et al. [30] also concluded that astrocytic lactate does not support neuronal activation. Most investigators, however, tend to test a new hypothesis using their own model system, whether in vivo or in vitro, and when the resulted outcome does not support the hypothesis, it is thrown by the wayside. Nonetheless, since they first proposed the ANLS hypothesis, the Magistretti group has produced several studies to support it [31,32,33,34,35,36,37,38,39,40], as did several other research groups [41,42,43,44,45,46,47,48,49,50,51,52,53,54,55,56,57]. Despite the growing number of studies that support the ANLS hypothesis and the role of lactate as the aerobic energy substrate during neuronal activity, the brain energy metabolism research community has remained divided over which molecule, glucose or lactate, is more important; glycolysis, meanwhile, is still split into aerobic and anaerobic pathways.

## 3. Methodological Limitations Can Lead to Miscalculations and Wrong Conclusions

An important issue that continues to raise doubts about the ANLS hypothesis and the oxidative use of lactate by activated neurons has to do with the way oxygen utilization is measured in vivo. Since hemoglobin in red blood cells is the carrier of oxygen, all measurements of oxygen utilization have been based on methods that could detect changes in the level of hemoglobin oxygenation or deoxygenation. In the brain, measurement of oxygen and glucose rates of consumption are known as the cerebral metabolic rate of oxygen (CMRO_2_) and of glucose (CMRglucose). Even the most advanced technologies today for measuring CMRO_2_, the different variations of magnetic resonance imaging (MRI) techniques, are based on measuring the changes in the level of deoxyhemoglobin, i.e., the level of oxygen extracted from the blood tissue [58]. Under ideal conditions, one would expect the ratio CMRO_2_/CMRglucose to be 6/1. Another way to measure how much oxygen is used during glucose breakdown is to measure the amount of CO_2_ released, where ideally, the ratio glucose/CO_2_ should be 1/1. The 6/1 ratio of CMRO_2_/CMRglucose or the 1/1 ratio of glucose/CO_2_ are never observed when the different measurement techniques are employed. This reality is best summarized by a quote from Saavedra-Torres et al. [59] (2021):

“…simultaneous measurements of glucose and oxygen consumption during rest or activation supposedly produces accurate estimate of the energy needs for the brain region under observation. However, the ratio CMRO_2_/CMRglucose values calculated are often significantly smaller than the expected 6/1. Such discrepancies have attributed to other glucose-consuming reactions not accompanies with oxygen consumption. Consequently, it has been a common understanding that a value of CMRO_2_/CMRglucose < 6 indicates that a partial non-oxidative glucose consumption (occurs). The smaller the value of CMRO_2_/CMRglucose, the greater is the non-oxidative consumption of glucose. This understanding makes sense when one assumes that a fully coupled glycolytic-mitochondrial respiratory apparatus should produce a CMRO_2_/CMRglucose value of 6 and an uncoupled apparatus (non-oxidative) should produce a CMRO_2_/CMRglucose value of ~0.”

The key word in the above quote is “*supposedly*”, regarding the simultaneous measurements of glucose and oxygen consumption and their accuracy. Assuming that glucose consumption measurements are accurate, measuring oxygen consumption by relying solely on changes in the level of oxygen in blood tissue (deoxyhemoglobin) could lead to errors, miscalculation, and the wrong conclusions, resulting in the misunderstanding of cerebral energy metabolism. As a segue to clarify the last sentence, a quote from the same publication mentioned above [59] is helpful:

“One of the most reliable techniques to measure oxygen consumption is the polarographic technique, which allows the determination of oxygen concentration via the measurement of the partial oxygen pressure (PO2) locally. Continuous measurements over a period of time when brain activity (EEG) is monitored demonstrated a correlation between increased activity and decreased tissue oxygen level. The development of oxygen microelectrodes has afforded a more accurate localization of such measurements.”

When one relies on blood oxygenation as the best way to determine the level of oxygen in any given brain region under study, one ignores the fact that oxygen exists outside the vasculature and in the brain tissue parenchyma itself. Therefore, the best technique to measure oxygen level is the polarographic one. Although this technique is not yet available for use in the human brain, it has been used successfully in the animal brain.

To understand how certain techniques could lead to “new discoveries”, the study by Fox et al. [60] is a good example. Published in the Journal *Science*, the paper’s title is “*Nonoxidative Glucose Consumption During Focal Physiologic Neural Activity*”. As expected, the paper raised a commotion among brain energy metabolism researchers, with the general understanding at the time being that brain activity demands and consumes great deal of energy, energy that can only be provided by “*fully coupled glycolytic-mitochondrial respiratory apparatus*”. The abstract to the paper by Fox et al. [60] succinctly describes their findings:

“Brain glucose uptake, oxygen metabolism, and blood flow in humans were measured with positron emission tomography, and a resting-state molar ratio of oxygen to glucose consumption of 4.1:1 was obtained. Physiological neural activity, however, increased glucose uptake and blood flow much more (51 and 50 percent, respectively) than oxygen consumption (5 percent) and produced a molar ratio for the increases of 0.4:1. Transient increases in neural activity cause a tissue uptake of glucose in excess of that consumed by oxidative metabolism, acutely consume much less energy than previously believed, and regulate local blood flow for purposes other than oxidative metabolism.”

The discovery that brain activity does not require oxygen, although surprising, has been accepted by most researchers as a fact. This pushed many to re-evaluate the role of two energy-producing processes, cytosolic glycolysis and mitochondrial oxidative phosphorylation, in brain activity. Understandably, glycolysis, the inefficient pathway by which a significantly smaller amount of ATP is produced, compared to that produced by oxidative phosphorylation, has become a focus of attention, mainly due to the findings of Fox et al. [60], which indicated that the energy necessary for brain activity does not involve consumption but only glucose, i.e., aerobic glycolysis. Everyone seemed to accept both the techniques used by Fox et al. [60], their findings, and their use of the term ‘aerobic glycolysis’, making it part of the established brain energy metabolism jargon. It is important to mention here that the term ‘aerobic glycolysis’ has been used for over a century to describe the process in which cancerous cells produce energy by converting glucose to lactate in the presence of oxygen, known as the Warburg effect [61]. However, excluding energy production of cancerous cells, aerobic glycolysis, according to its original concept and the way most scientists understand it, is the first stage that leads to the major respiration pathway, i.e., the mitochondrial TCA cycle and its coupled oxidative phosphorylation, which cannot proceed without oxygen. And yet, one study in 1988 [60] completely changed the meaning of the term ‘aerobic glycolysis’ to mean ‘glycolysis in the presence of oxygen and mitochondria that ends with lactate and occur normally in the active brain.’ This contradiction seems to be less bothersome to most than the ability of the brain to utilize lactate oxidatively or the ANLS hypothesis. Ironically, aerobic glycolysis, with its new meaning, makes both lactate oxidative utilization and the ANLS hypothesis logical. Therefore, one must question not the results reported by Fox et al. [60], but rather their interpretation. Since they used the level of blood oxygenation as the parameter by which oxygen consumption is calculated, they had to assume that all the oxygen that active neural tissue consumes is stored in the blood compartment and nowhere else. Because they could not observe any change in blood oxygenation during neural activation, a conclusion had to be made that the neural activity is supported by nonoxidative glucose consumption.

Following up on the quote by Saavedra-Torres et al. [59] above, regarding the accuracy of the polarographic technique for the measurement of oxygen consumption, Hu and Wilson [41] designed and used a microelectrode to measure oxygen fluctuations in the rat hippocampus in vivo, where the electrode was placed in parenchyma and not in the blood vasculature. Hu and Wilson also simultaneously measured glucose and lactate in the same brain region using micro sensors designed to detect these molecules. Their results, analyzed [16] and reanalyzed [62], clearly showed that neuronal activity in the rat hippocampus, induced by electrical stimulation, is supported by oxidative utilization of lactate. Both oxygen, consumed during that activation, and glucose were present in the tissue and the latter replenished the consumed lactate during the stimulation. Hu and Wilson [41] summarized their findings as follows:

“The results of simultaneous measurements of local extracellular concentrations of lactate, glucose, and O_2_ first provide a comprehensive picture of complementary supply and use of lactate and glucose behind the BBB. In response to acute neuronal activation, the brain tissue shifts immediately to significant energy supply by lactate. A local temporary fuel pool behind the BBB is established by increased extracellular lactate concentration. The glutamate-stimulated astrocytic glycolysis and increase of regional blood flow may regulate the lactate concentration of the pool at different levels (i.e., molecular/cell level and anatomic level, respectively), and on different time scales, to maintain local energy homeostasis. Thus, the formation of a temporary local extracellular fuel reservoir resolves nicely the conflict between the demands of local energy homeostasis and the transport from the circulatory system.”

Hu and Wilson [41] elegantly tied together the original finding that neuronal function can be solely supported by the oxidative utilization of lactate [2], the “nonoxidative glucose consumption” during neural activation [60], and the ANLS hypothesis [19]. Aerobic glycolysis (nonoxidative glucose utilization), in today’s jargon, is lactate production in the presence of oxygen, signifying a glycolytic ATP production that bypasses the mitochondrial TCA cycle and oxidative phosphorylation. However, based on the findings of Hu and Wilson [41], there is clearly sufficient oxygen present in brain tissue outside the vasculature to support lactate oxidative utilization. There is also sufficient lactate in the tissue, which could easily be supplemented, if necessary, by glucose (glycolysis). The hippocampal activity, stimulated electrically in the study by Hu and Wilson [41], is glutamatergic in nature, and the astrocytic glutamate uptake would be responsible for the glucose uptake and the production of lactate (the ANLS hypothesis). Moreover, this summation allows one to conclude that glycolysis, in the brain and elsewhere, always ends up with lactate [12] and that the terms ‘aerobic’ and ‘anaerobic’ are unnecessary and misleading [11,16,51,62,63,64,65,66,67]. Inexplicably, the study by Hu and Wilson [41] never received the attention it deserves (382 citations), especially when one compares it to the attention the paper by Fox et al. [60] received (2269 citations). Such a discrepancy cannot be explained as healthy scientific skepticism. Those who accept the concept of *nonoxidative glucose utilization by the active brain* also support the concept of two glycolytic pathways, aerobic and anaerobic; a real paradox that has already created great confusion. Consequently, by holding to the nonoxidative glucose utilization concept, its supporters have offered a hypothesis to resolve this paradox, naming it *the efficiency tradeoff hypothesis*, which was supposed to account for the observed results that emerged from the use of the BOLD fMRI technique [68]. The validity of the efficiency tradeoff hypothesis should be questioned, since it relies on the BOLD fMRI signal, a technique that measures deoxyhemoglobin and not the consumption of oxygen presents in the brain tissue itself, which occurs instantaneously upon stimulation, as illustrated by Hu and Wilson [41]. Considering that the stimulated neural tissue is saturated with enough oxygen to answer the needs for oxidative metabolism of glucose (or lactate), measuring delayed oxygen extraction from the blood (BOLD signal) cannot accurately estimate the amount of oxygen consumed by the stimulated tissue. Interestingly, Theriault et al. [68] themselves recognized the multiple drawbacks that the BOLD signal may introduce to the calculation of oxygen consumption. Yet, they argued that lactate production during brain stimulation supports the idea of ‘aerobic glycolysis.’ Alternatively, if lactate is always the end-product of glycolysis, including during respiration, and if its production upon stimulation exceeds the levels consumed by mitochondria for oxidative phosphorylation [16,67], then the tracing of lactate efflux by Theriault et al. [68] should be the expected scenario, as indicated by Hu and Wilson [41]. Therefore, one could argue that an estimation of the rate of cerebral oxygen utilization via measurement of deoxyhemoglobin level (BOLD) is the wrong approach, as it may underestimate the rate of oxygen consumption and may lead to the wrong conclusion. One such conclusion is the nonoxidative glucose utilization (aerobic glycolysis) in response to neural activation, since it estimates oxygen consumption to be nil.

Regardless of these considerations, many scientists in the field of brain energy metabolism continue to rely on the BOLD fMRI signal as the best measurement of oxygen consumption. Consequently, they support the concept of aerobic glycolysis (nonoxidative glucose utilization) as the energy metabolic process that supports neural activity. To justify an inefficient process such as glycolysis, where ATP production is concerned, Theriault et al. [68] hypothesized that a faster process is needed to provide the required energy for neural activation and therefore named it *the efficiency tradeoff hypothesis*. If one considers the first stage of energy production from glucose (glycolysis) to be faster than the full process of respiration, which includes the multi-reactions of the mitochondrial TCA cycle and its coupled oxidative phosphorylation, then yes, the glycolytic pathway, with its 11 reactions, is faster than the full process of respiration. However, if the real process that supports neural activity is the mitochondrial oxidative utilization of lactate [11,12,16,41,43,51,62,63,65,66,67], a process that bypasses glycolysis altogether, then one could argue that this glycolysis-less route is almost as fast and much more efficient than glycolysis alone. As has been suggested elsewhere, “glycolysis” should carry no prefix, neither “aerobic” nor “anaerobic” [66], since it is a pathway that always proceeds to completion, i.e., the production of lactate, independent of the presence or absence of oxygen and/or mitochondria.

## 4. Scientific Bias May Influence the Research of Brain Activation and Its Energy Needs

Why is there such a strong willingness to accept an inefficient, fast process over an alternative one that is significantly more efficient and not necessarily slower as the process that fuels neural activity? Here, a habit of mind cannot fully explain such willingness, as even the most ardent supporters of the role of glucose as the obligatory brain energy substrate are now accepting lactate as a player in energy metabolism of the brain and other organs. One should hesitate to use the word ‘bias’ to describe the reason for this willingness. However, the tendency to minimize the role and importance of lactate in brain energy metabolism appears to be driven mainly by Louis Sokoloff’s students and postdoctoral fellows who hold to the following mantra:

“That oxygen and glucose consumption and carbon dioxide production are essentially in stochiometric balance and no other energy laten substrate is taken from the blood means, however that the net energy made available to the brain must ultimately be derived from the oxidation of glucose. It should be noted that this is the situation in the normal state; … other substrates maybe used in special circumstances or in abnormal states.” [69].

Of course, a great deal of information about the brain and its energy metabolic machinery has been gained over the three decades since the above paragraph was written. Nevertheless, Sokoloff’s followers regularly minimize, reject, or just overlook that information and find themselves categorizing brain activation as a special circumstance or abnormal state to explain nonoxidative glucose utilization (aerobic glycolysis) and lactate production. Any of these three deeds could be described as a scientific bias.

In his book, *Science Fictions: Exposing Fraud, Bias, Negligence and Hype in Science*, Stuart Ritchie [70] describes the type of bias in science, where a hypothesis is shared by a community of scientists who also shared their bias regarding said hypothesis. When among the members of such a group there are “powerful, well-established professors,” their common bias could express itself by exhibiting strong objection to any hypotheses or findings that disagree with the group’s hypothesis. Where the two hypotheses being discussed here are concerned, *the efficiency tradeoff hypothesis* is the latest salvo the disciples of Sokoloff’s used [68] in their attempt to discredit *the ANLS hypothesis*. The list below (Table 1) summarizes these attempts chronologically. 

In 26 scientific papers that were authored or co-authored by Sokoloff’s followers, the principal message aimed at questioning or completely dismissing the ANLS hypothesis, spurred by the collective bias of the group against the role of lactate in brain activation. That bias began in the laboratory of Louis Sokoloff and spread by his disciples. There were 6 out of these 26 papers that cited the publication of Hu and Wilson [41], all with the same message, declaring that “*contributions of changes in extracellular glucose and lactate levels to total metabolism were overestimated by Hu and Wilson because they did not take into account the metabolic through-flux of blood-born glucose*”. However, the analysis of Hu and Wilson’s results [16,62] clearly indicated that lactate did become the main oxidative energy substrate to produce ATP during stimulation. Moreover, lactate shown to spare glucose utilization post stimulation. This kind of bias, similarly to habit of mind, impedes the progress in the field of brain energy metabolism [66]. Listed below are the most telling statements of bias against the functions and fate of lactate in the brain as they appear in the multi-authored review [93]:


*There has been a wealth of papers over the past few decades on lactate metabolism and the role(s) of lactate in the brain, with some 16,000 papers resulting from a search for lactate metabolism AND brain in PubMed. Despite all this investigation, lactate is likely the most controversial and misunderstood brain energy substrate.*


Interestingly, lactate is recognized as a brain energy substrate despite the bias.


*Lactate is formed in all brain cells solely from pyruvate in a reversible reaction catalyzed by lactate dehydrogenase, requiring NADH as the cofactor. Lactate dehydrogenase is a ubiquitous near-equilibrium enzyme, which means it has limited capacity to influence metabolic control. While lactate itself is a “dead-end” metabolite, its immediate substrate, pyruvate, is a key molecule at a metabolic crossroad. Pyruvate can be generated by oxidative metabolism of glucose, glutamate, and other substrates via the pyruvate recycling pathway. It is converted to acetyl-CoA via the far-equilibrium pyruvate dehydrogenase complex and can also be transaminated to alanine in a pairing with glutamate and 2-oxoglutarate or carboxylated to form oxaloacetate by pyruvate carboxylase, a glial-specific enzyme. This plethora of possible pathways for pyruvate, and their respective thermodynamics, means that pyruvate clearance rates (i.e., substrate availability) tend to have more control over production of lactate than lactate dehydrogenase itself. Lactate (dehydrogenase) isoenzymes, while able to influence the rate (time course) of conversion of substrates, have no influence on the equilibrium lactate concentration.*


The authors hold to the old mantra that lactate is a product of pyruvate oxidation in a reversible equilibrium reaction by the enzyme lactate dehydrogenase (LDH). They ignore the evidence provided by many studies of the existence of a separate mitochondrial LDH that reduces lactate to pyruvate, while the cytosolic LDH is the one that oxidizes pyruvate to lactate. The concept that this enzyme also catalyzes the reverse reaction is as archaic as the concept of two types of glycolysis, aerobic and anaerobic [12,51,66,95,96,97,98,99,100].


*Another important, and often forgotten factor, is that full lactate utilization in the respiratory chain requires oxygen (CMRO_2_), and if oxygen consumption does not match glucose utilization, then the lactate derived from glucose cannot be oxidized. During activation, glycolysis is preferentially upregulated compared with oxidation. Specific activity measurements indicate that most of the lactate produced from glucose via this upregulation is retained and oxidized in the brain. Specific activity measurements also indicate that lactate derived from glycogen in astrocytes is quickly released from the brain because, if retained, lactate-specific activity would be diluted. Lactate may escape from its “dead-end” reaction by leaving the cell, which it does through an array of transporters which mediate the uptake and release of lactate and other monocarboxylates such as acetate, pyruvate, and the ketone bodies β-hydroxybutyrate and acetoacetate. Lactate may also escape via gap junctions where it travels relatively long distances.*


As mentioned earlier, reliance on the level of deoxyhemoglobin measurements for estimates of oxygen consumption is inaccurate and may lead to incorrect conclusions. As shown by Hu and Wilson [41], most of the oxygen consumed during brain activation in vivo already exists in the parenchyma, outside the vasculature, and is readily available for lactate oxidative utilization and that glucose utilization is minimal [16,62,67]. Moreover, an important study by Li et al. [101], who used ultrasensitive and highly responsive lactate sensors, has demonstrated the presence of a lactate pool inside mammalian mitochondria, and that the lactate level in the mitochondria is significantly higher than that in the cytosol or nucleus. Both studies, the one by Hu and Wilson [41], and the more recent one by Li et al. [101] were not considered at all in the review article by Rae et al. [93]. While recognizing lactate is a brain energy substrate (statement #1), the authors determined that its utilization requires oxygen. And since they also concluded that BOLD measurements indicate that oxygen is not consumed (aerobic glycolysis), they determined that the lactate produced reaches a ‘dead-end’, and the only way to escape is to exit the cell via its own monocarboxylate transporters. Again, none of that really takes place according to the findings of Hu and Wilson [41], Li et al. [101], and all the other published papers cited in the response to statement #2.


*Lactate production in the human brain appears to be region-specific in a pattern that is conserved across individuals, with production highest in the parietal and occipital lobes (cuneus, precuneus, cingulate, and lingual gyri). Production of lactate is also not necessarily related to regional tendency to aerobic glycolysis or FDG uptake, although it may be related to the availability of NADH. Exactly when and why the brain produces lactate is still not well-understood, although many possible explanations have been advanced. It has been speculated that the need to rapidly clear glutamate from the brain is fueled by glycolysis, producing pyruvate which is excess to requirements. The excess pyruvate is then effluxed as lactate, regenerating cytosolic NAD^+^. Alternatively, the controversial astrocyte–lactate shuttle hypothesis posits that lactate is released by astrocytes in response to glutamate release from neurons, where the lactate is taken up and used as a fuel. Many groups have embraced the concept of an astrocyte to neuron lactate shuttle (ANLSH/ANLS) since it was first proposed by Pellerin and Magistretti based on in vitro studies. However, studies in vivo supporting this shuttle are few and not convincing. The thermodynamics of lactate metabolism, transport, and clearance go some way to explaining the attraction of the ANLS. As the reactions involving lactate are all near-equilibrium enzymes, lactate is highly correlated with many metabolites and processes, which gives lactate a seeming importance beyond the reality. This was explained succinctly by the renowned biochemist, Richard Veech, who stated that “measurements of lactate content per se are able to provide relatively little information other than the level of lactate itself. By combining the measurement of lactate with the concentration of its relevant metabolic partners, a great deal more useful information can be gained about the state of the tissue.” So is there an astrocyte–neuron lactate shuttle? Sometimes, under the right conditions but it is probably just lactate exchanging between compartments. Is it important? Well, it can happen, but in the grand scheme of things, no, it is not that important, as lactate is just as likely to efflux from neurons and is not a substrate that can maintain high level neuronal function. Giving it a name (ANLS) has given it an importance, which outstrips its role as one of many reasons that lactate enters or leaves cells and the brain.*


Here, the authors claimed that not much is known about lactate production and its utilization and suggested that as the brain prevents itself from following lactate production by aerobic glycolysis, it could benefit by the regeneration of NAD^+^. Simultaneously, they depreciated the alternative explanation that the ANLS hypothesis provides, concluding that it is mainly its name, ANLS, which provided it with an undeserved importance.

Either because of bias against lactate and its role and function in brain activation, or bias in favor of glucose as the obligatory do-it-all molecule, much effort, time, and resources are being spent to disprove a working hypothesis (ANLS), while advancing another that is based on a questionable measurement technique (BOLD fMRI).

The ANLS hypothesis is not the only working concept that established, and later demonstrated through many studies, the use of lactate as the preferred neuronal oxidative substrate during brain activity. Only a sample of the increasing number of both human and animal studies showing brain utilization of lactate that originates from muscle and other tissues is cited here [102,103,104,105,106,107,108,109,110].

## 5. A Unifying Concept of Brain Activation and the Energy Metabolic Pathways That Support It

The two competing hypotheses addressed here, the astrocyte–neuron lactate shuttle (ANLS) hypothesis and the tradeoff efficiency hypothesis, are illustrated in Figure 1 below. The latter was hypothesized to explain a perplexing observation made first by Fox et al. [60], where brain activity is supported by glucose consumption that was not accompanied by oxygen consumption. Since then, this observation has been reproduced countless times, including when the most advanced technique (BOLD fMRI) was used to estimate oxygen consumption, and therefore should be indisputable. This estimate is based on a signal produced by deoxyhemoglobin in red blood cells carried by brain blood capillaries. When the BOLD signal indicates that the level of deoxyhemoglobin at the measuring site did not increase upon brain activation, it is interpreted to mean that oxygen was not consumed. However, when one relies on BOLD measurements and estimations, one does not take into consideration the possibility that oxygen is present outside blood vessels and at sufficient levels. Extra-vasculature oxygen could be used instantly to fulfill all the energy needs of the active brain via lactate utilization in the mitochondrial TCA cycle and its coupled oxidative phosphorylation, conditions that the ANLS hypothesis posits (Figure 1A). According to the elegant and accurate measurements performed in vivo by Hu and Wilson [41], the oxygen present in the tissue outside the vasculature is abundant and readily consumed during glutamatergic stimulation in complete concert with lactate consumption. These measurements and their analyses [16,41,62] fit well with the energy metabolism processes of the activated brain as described by the ANLS hypothesis. The tradeoff efficiency hypothesis (Figure 1B), postulating that oxygen does not play a part in producing the necessary energy for brain activity, is therefore based on a wrong concept.

In a general sense, it would be more fruitful for the brain energy metabolism research community to do away with the old, dogmatic concept of dividing the glycolytic pathway into aerobic and anaerobic types [11,16,51,62,63,64,65,66,67]. The sooner we accept the concept of a single glycolytic pathway that begins with glucose and, through 11 enzymatic reactions, ends with lactate, independent of oxygen and/or mitochondria, the earlier we achieve a better understanding of its participation in the process of brain energy metabolism in rest and during activity. Only then would we be able to consider whether glycolysis is, by itself, the answer to the energy needs of the active brain, or if its end-product, lactate, that is the real substrate for a fast and efficient mitochondrial oxidative phosphorylation. Both oxygen and lactate levels in brain tissue are sufficient to support stimulated brain activity, while glucose continues to supplement lactate when it is needed [16,41,62]. All the intra- and extracellular machinery required to maintain the utilization of lactate as the substrate of mitochondrial oxidative phosphorylation is included in the ANLS proposal. In astrocytes, the glycolytic pathway supports the Na^+^/K^+^ ATPase, which is induced by glutamate uptake, and the lactate produced thereof is transported via an array of monocarboxylate transporters (MCT1, MCT2) out of astrocytes and into neurons (ANLS). In neurons, it is shuttled into mitochondria as the substrate of the TCA cycle, where it is being oxidized into pyruvate by the mitochondrial lactate dehydrogenase (mLDH). In short, they are cell-to-cell and intracellular lactate shuttles [45]. Moreover, since glycolysis always ends up with lactate, its cyclical nature assures the production of NAD^+^, a function that would not be possible if glycolysis ended with pyruvate. Similarly, astrocytic lactate is utilized by astrocytes themselves as the substrate of the mitochondrial oxidative phosphorylation. All around, these processes are both efficient and sufficient to answer all the energetic needs of the active brain.

## 6. Coda

For over eight decades, the glycolytic pathway has been presented as having two different outcomes that are, depending on the presence or absence of oxygen, aerobic or anaerobic, respectively. The continuing teaching and researching of brain energy metabolism, using pyruvate and lactate as the two separate products of glycolysis, had forced the investigators of brain energy metabolism to hypothesize that glycolysis in the presence of oxygen could end with the production of lactate, naming it ‘aerobic glycolysis.’ This aerobic glycolysis cannot be differentiated from the original pathway that ends with pyruvate. That is despite red blood cells being known to produce energy only via glycolysis that ends with lactate even in the presence of ample oxygen, or that the glycolytic energy production of cancerous cells is also tagged ‘aerobic glycolysis [61]. All this confusion could be avoided if glycolysis was accepted as a single pathway of 11 enzymatic reactions that break down glucose to lactate, independent of the presence or absence of oxygen. For this new paradigm to be accepted, many must overcome their **habit of mind**, the barrier that prevents them from acceding to the necessary paradigm shift [11,17,65,66]. Others will have to rid themselves of any scientific bias that does not allow them to consider lactate to be more than a small player in brain energy metabolism, both in rest and during activity. Lactate is the substrate of the mitochondrial TCA cycle, and whenever the active brain requires a substrate to produce ATP, the choice will be lactate over glucose if there is enough supply of the monocarboxylate. When that supply is diminished, glucose is hydrolyzed via the glycolytic pathway to renew it [41]. Therefore, as things stand today in this field of research, the astrocyte–neuron lactate shuttle (ANLS) hypothesis is the best available explanation of the numerous observations, measurements, calculations, and estimations made both in vivo and in vitro when researching brain energy metabolism in rest and during activity. Certain techniques, such as BOLD fMRI, despite their accuracy and clarity of details they provide, have their own drawbacks. Specifically, where this technique is concerned, although it can detect minute changes in oxygen content within blood vessels, it does not detect oxygen content of brain tissue outside the blood vessels. The reliance of many on this technique has led to the formulation of the far-fetched efficiency tradeoff hypothesis supported by respected scientists twisting their brains to explain a reading of a signal (BOLD) that cannot detect any consumption of oxygen outside of blood vessels. However, oxygen electrodes, unlike the BOLD signal, sense and measure tissue oxygen fluctuations directly. The active brain is the one organ that consumes more energy than any other organ in the body. Without being as efficient as possible, the active brain will cease its activity. The skepticism over the validity of the ANLS hypothesis aside, this opinion paper’s principal purpose, as its title indicates, is to underscore the lactate’s role in brain energy metabolism. Accordingly, lactate, not pyruvate, is always the glycolytic end-product, independent of the presence or absence of oxygen or mitochondria, while also being the real oxidative energy substate of the mitochondrial TCA cycle.

## Figures and Tables

**Figure 1 ijms-26-04429-f001:**
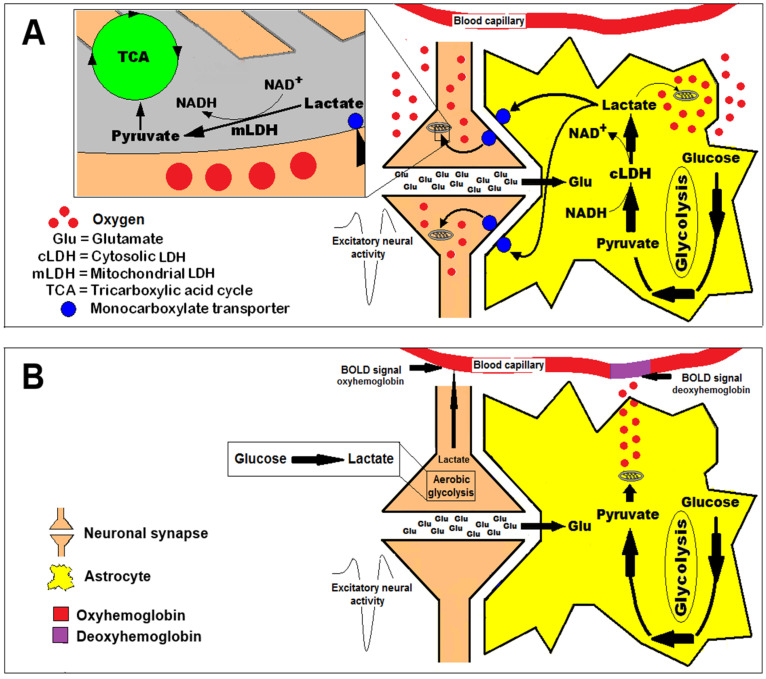
A schematic presentation of the two hypotheses offered to explain how energy metabolism supports brain activation. (**A**) The astrocyte–neuron lactate shuttle (ANLS) hypothesis [19], suggesting that the energy substrate, lactate, which supports neural activity, originates from astrocytic glycolysis that is induced by glutamate uptake and activation of the Na^+^/K^+^ ATPase pump. The astrocytic lactate produced is shuttled via monocarboxylate transporters to the active neuron where it is used oxidatively (mitochondrially) in support of the neural activity. Hu and Wilson [41] showed that lactate is the main substrate used during neural stimulation and that the oxygen necessary for its utilization is present in the tissue parenchyma outside of the blood vasculature and is plentiful. (**B**) The efficiency tradeoff hypothesis [68] suggests that aerobic glycolysis (nonoxidative glucose consumption and lactate production) is the process that supplies the active neural tissue with the necessary energy. Although glycolysis is an inefficient metabolic pathway, the hypothesis claims that the speed by which glycolysis produces ATP, as compared to the slower oxidative phosphorylation, trades speed for efficiency, which is necessary for the support of brain activation. This hypothesis was constructed to explain the lack of oxygen consumption detected when the BOLD fMRI technique was used during brain activation. The technique measures the level of deoxyhemoglobin in red blood cells carried by blood capillaries, and is not the direct measurement of tissue oxygen that an oxygen electrode affords. The efficient tradeoff hypothesis still contends that under aerobic conditions at rest, pyruvate is the glycolysis end-product.

**Table 1 ijms-26-04429-t001:** A chronological list of publications by scientists from the Sokoloff school, including a short synopsis of each publication. This list exemplifies the concerted efforts made by these scientists for almost three decades to reduce or nullify the importance of lactate as an oxidative energy substrate. To do so, they question the validity of the ANLS hypothesis, which indicates neuronal utilization of astrocytic-produced lactate during brain activation. While there are other studies from different laboratories that do question the importance of ANLS in brain activation, they do not necessarily question the role of lactate as an energy substrate. Nevertheless, at least 28 studies are cited [24,25,31,32,33,34,35,36,37,38,39,40,41,42,43,44,45,46,47,48,49,50,51,52,53,54,55,56,57] in this paper that support the ANLS hypothesis.

1. Takahashi et al. [71]. The first publication from Sokoloff’s laboratory, citing the ANLS hypothesis and agreeing with the observation that brain activation stimulates astrocytic glucose consumption. 2. Sokoloff et al. [20]. Another publication that agrees with the participation of astrocytes in increased glucose metabolism upon stimulation. 3. Sokoloff [72]. A paper summarizing his laboratory’s findings about astrocytic ion uptake and glucose consumption, without mentioning the other portion of the ANLS hypothesis, which is the neuronal oxidative consumption of lactate. 4. Cruz et al. [73]. The first paper from this school that relates to the lactate produced during brain activation, limiting the study to the brain cortex and only measuring lactate efflux to blood and estimates of efflux to other brain regions. A possible lactate oxidative utilization was not considered. 5. Dienel and Hertz [21]. A publication by one of Sokoloff’s disciples, who combined forces with Professor Leif Herz, an opponent of the ANLS hypothesis, in minimizing the glycolytic production of lactate to normal overflow upon brain activation, while arguing that glutamate uptake by astrocytes is supported by oxidative metabolism, not aerobic glycolysis. 6. Hertz and Dienel [74]. In a review article, the authors summarized the central role of glucose in energy metabolism, while also discounting all in vitro systems used in the study of brain energy metabolism. They focused on the use of MRI as the technique that allows the measuring of local glucose metabolism in vivo and thereby confirming many of the in vitro findings, while also ignoring any role for lactate as an oxidative energy substrate for neural activity. 7. Dienel and Cruz [75]. The first paper by Dienel and Co. to argue specifically against the ANLS hypothesis. 8. Hertz [76]. A paper that echoes Dienel and Cruz’s (2004) argument, i.e., that there is “no solid indication that lactate oxidation should primarily occur in neurons, and there is very good evidence that it is not exclusively a neuronal process”. 9. Dienel [77]. In an Editorial, the author, although accepting of the fact that lactate could be an oxidative substrate in the brain, continued to argue against the ANLS hypothesis. 10. Sokoloff [78]. In this book chapter, Sokoloff himself seemed to accept the possibility of an astrocytic–neuronal lactate shuttle and lactate oxidative utilization. 11. Hertz and Dienel [79]. Accepting the fact that lactate can be used oxidatively by brain cells, the authors argued that the neuronal lactate transporter (MCT2) is a high-affinity transporter that would not allow for a speedy uptake of lactate necessary for immediate use. 12. Dienel and Hertz [80]. The results presented in this paper are very similar to those found in the paper by Hertz and Dienel (2005). 13. Abe et al. [81]. This study found that certain astroglia, upon activation of the Na^+^/K^+^-ATPase, prefer lactate oxidation over glucose. One of the authors is S. Takahashi, who worked in Sokoloff’s laboratory. 14. Dienel and Cruze [82]. The authors tested the ability of astrocytes in vitro to utilize substrates other than glucose during activation. They tested glycogen, acetate, and glutamate, concluding that all three could be utilized by astrocytes during stimulation. Interestingly, they did not test lactate (why?). 15. Hertz et al. [83]. In this review article, the authors summarized their findings as follows: Oxidative metabolism, glycolysis, and, in some cases, glycogenolysis in astrocytes, increase when their workload rises. The energy requirement for glutamate uptake and metabolism to glutamine may be the most generally recognized energy-requiring process in astrocytes. During brain stimulation, active K^+^ uptake may pose the largest demand on ATP and especially on glycolytically generated ATP. Energy balance sheets and models of astrocyte–neuron interactions that do not include these functions greatly underestimate astrocytic energetics and portray a very limited context within which to understand astrocytic energetics. When lactate is produced during these processes, its fate is unknown. 16. Cruz et al. [84]. Using ^14^C-labled glucose and 2-deoxyglucose, the authors concluded that upon brain activation, in vivo focal CMRglucose is underestimated when using labeled glucose “because of decarboxylation reactions, spreading within tissue and via the astrocyte syncytium, and release from activated tissue”. They stated that such underestimation could “explain the fall in CMRO_2_/CMRglucose during brain activation and suggest that lactate and other non-oxidized metabolites of glucose are quickly shuttled away from sites of functional activation”. The possibility that the fall could be due to oxidative utilization of lactate rather than glucose was not considered. 17. Gandhi et al. [85]. Another study from Dienel’s laboratory where labeled glucose was used to measure its utilization and lactate production, which insisted that lactate is slow to be taken up by neurons compared to the ability of astrocytes, and argued that most of the lactate efflux to the blood stream and to other brain regions. Lactate oxidative utilization by neurons was not considered. 18. Dienel [86]. Another review article where the author, after a myriad of studies by various groups that consistently showed lactate to be an oxidative neuronal substrate, relented somewhat: “Brain activation in subjects with low plasma lactate causes outward, brain-to-blood lactate gradients, and lactate is quickly released in substantial amounts. Lactate utilization by the adult brain increases during lactate infusions and strenuous exercise that markedly increase blood lactate levels. Lactate can be an ‘opportunistic’, glucose-sparing substrate when present in high amounts, but most evidence supports glucose as the major fuel for normal, activated brain”. 19. Dienel [87]. A second review article with the same conclusion: “Brain activation in subjects with low blood-lactate levels causes a brain-to-blood lactate gradient, with rapid lactate release. In contrast, lactate flooding of brain during physical activity or infusion provides an opportunistic, supplemental fuel. Available evidence indicates that lactate shuttling coupled to its local oxidation during activation is a small fraction of glucose oxidation”. 20. Dienel [88]. An invited review where the author made the following conclusion: “Three lines of evidence indicate that critical cornerstones of the astrocyte-to-neuron lactate shuttle model are not established, and normal brain does not need lactate as supplemental fuel”. 21. Dienel [89]. An opinion article that touched on the potential of treating traumatic brain injury with lactate: “Results show that lactate release from human brain to blood predominates over its uptake after TBI, and strong evidence for lactate metabolism is lacking; mitochondrial dysfunction may inhibit lactate oxidation. Claims that exogenous lactate infusion is energetically beneficial for TBI patients are not based on metabolic assays and data are incorrectly interpreted”. 22. Dienel [90]. Another review that downplayed the role of lactate in astrocytic–neuronal lactate shuttle: “Glucose is the obligatory fuel for adult brain, but lactate produced from glucose by astrocytes within brain during activation has been proposed to serve as neuronal fuel. However, metabolic requirements for substantial lactate shuttling and oxidation are not fulfilled, and this notion is refuted by several independent lines of evidence. Understanding preferential upregulation of glucose compared with oxygen utilization is central to elucidating brain energetics”. 23. Dienel [91]. A research paper that highlighted the following: “Glucose and lactate enhance memory; glycogen is required for its consolidation. Hogh-dose lactate preserves memory and gene expression without glycogenolysis. High-dose lactate suppresses neuronal firing and is taken up mainly by astrocytes”. 24. Dienel [92]. Review article that concluded: “Shuttling of glucose- and glycogen-derived lactate from astrocytes to neurons during activation, neurotransmission, and memory consolidation are controversial topics for which alternative mechanisms are proposed”. 25. Rae et al. [93]. A collaboration of Dienel with 25 other authors on a review article that held to the dogmatic concept, according to which under resting conditions glycolysis ends with pyruvate, not with lactate. When lactate is produced (aerobic glycolysis), it “may escape from its “dead-end” reaction by leaving the cell, which it does through an array of transporters”. Where the ANLS hypothesis was concerned, this review stated: “Many groups have embraced the concept of an astrocyte to neuron lactate shuttle since it was first proposed … based on in vitro studies. However, studies in vivo supporting this shuttle are few and not convincing”. 26. Dienel and Rothman [94]. In this research paper, the authors used a genetically encoded lactate biosensor to measure changes in the level of the monocarboxylate both in vitro and in vivo. The impetus for this initiative was several studies that showed a gradient of lactate from astrocytes to neurons. The authors concluded that unless a calibration of the biosensor is performed both for the metabolism of lactate during the measurement and for the cellular volume of the cell-type under monitoring, any measurement is questionable.

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
