# Peer review of "The Feud over Lactate and Its Role in Brain Energy Metabolism: An Unnecessary Burden on Research and the Scientists Who Practice It"

_ijms, 2025, doi:10.3390/ijms26094429_

Round 1
Reviewer 1 Report
Comments and Suggestions for Authors
I have enjoyed a lot the reading of this manuscript, in particular the early history of lactate, which I only knew it in part.
This manuscript revise thoroughly earlier literature on lactate, trying to bridge a gap between apperently confusing results in favor or against ANLS. To this regard, I have a couple of suggestions:
- Table 1 do not cite any of Magistretti's paper. I think at least 1994 PNAS (Pellerin and Magistretti) and Suzuki et al. 2011 (Cell) would deserve to be included. Although the review is much more on the metabolic detail of Lactate, I believe that the functional relevance is still one of the key points that was convincing for the scientific community.
- Regarding the methodological limitations, might be worth mentioning that works showing the presence of glycogen nearby synapses, like Calì et al., 2016 (J comp neurol), Vezzoli et al., (2020), Bellesi et al., (2015), pointing to the fact that lactate utilization might happen in subdomains at a distance from the blood vessels and with a timing that is well below BOLD dectection.
Author Response
- Table 1 do not cite any of Magistretti's paper. I think at least 1994 PNAS (Pellerin and Magistretti) and Suzuki et al. 2011 (Cell) would deserve to be included. Although the review is much more on the metabolic detail of Lactate, I believe that the functional relevance is still one of the key points that was convincing for the scientific community.
- Regarding the methodological limitations, might be worth mentioning that works showing the presence of glycogen nearby synapses, like Calì et al., 2016 (J comp neurol), Vezzoli et al., (2020), Bellesi et al., (2015), pointing to the fact that lactate utilization might happen in subdomains at a distance from the blood vessels and with a timing that is well below BOLD dectection.
-
I thank the reviewer for the review and the points made. Table one has been constructed specifically to point out the concerted effort of one group of scientists, all graduates of the late professor Sokoloff’s NIH laboratory, who continue to carry on his mantra that there is no substitute for glucose, the obligatory energy substrate in the brain. Many of the studies (28 of them) that do support Pellerin and Magistretti’s ANLS hypothesis are mentioned in the text and listed in the list of references, including Pellerin and Magistretti’s 1994 publication. Although the reviewer is correct about glycogen as a source of lactate in the brain, its role, as many other roles lactate plays in multitude of brain functions, is beyond the scope of the present manuscript. Accordingly, the word “beyond” was removed from the title of the manuscript, since my aim is to specifically focus on lactate as always being the glycolytic end-product, independent of the presence or absence of oxygen or mitochondria, and the real oxidative energy substate of the mitochondrial TCA cycle.
Reviewer 2 Report
Comments and Suggestions for Authors
The article by Avital Schurr reviews the long-standing controversy regarding the role of lactate in brain energy metabolism, advocating that glycolysis always ends with lactate production, regardless of the presence of oxygen, and criticizing the traditional division between aerobic and anaerobic glycolysis. Schurr argues that lactate is the principal oxidative substrate for the active brain, as proposed by the Astrocyte-Neuron Lactate Shuttle (ANLS) hypothesis. He criticizes the reliance on techniques such as BOLD-fMRI, which measure only blood oxygenation and not tissue oxygen consumption, leading to the formulation of the Efficiency Tradeoff Hypothesis — considered by the author to be a misinterpretation. The manuscript also highlights what the author perceives as a scientific bias in favor of glucose as the mandatory brain fuel, which, according to him, has delayed the acceptance of lactate's role.
Although the topic addressed is relevant for understanding brain energy metabolism, the manuscript presents a strong opinionated bias, lack of scientific neutrality, absence of discussion of new experimental evidence and selective interpretation of the literature.
Major Comment:
The author demonstrates excessive bias throughout the text. He repeatedly disqualifies the scientific community that does not agree with his view on lactate. Instead of evaluating opposing evidence in a balanced manner, he attributes disagreements to mental habits and prejudices, which is an anti-scientific attitude.
The text frequently adopts a personal and emotional tone against other research groups. Terms such as "bias," "dogmatic," "twisting their brains," "scientific bias," and "unnecessary burden" appear repeatedly.
Although the manuscript correctly critiques the limitations of methods such as BOLD-fMRI, it does not deeply explore the legitimate reasons that have led scientists to question the ANLS hypothesis. References are selectively cited ("cherry-picking") to strengthen only the author's viewpoint.
The manuscript structure resembles more an opinion essay rather than a systematic review. There are no summary tables comparing supporting and opposing studies, nor a standardized critical analysis of the literature.
The author repeatedly cites his own previous works, which, although relevant, may convey a sense of self-affirmation bias.
Despite offering valuable contributions to the appreciation of lactate as an energy substrate, the article suffers from overconfidence in a single model (ANLS), underestimates methodological advances, and fails to recognize the metabolic plasticity of the brain. A more balanced approach, considering different physiological scenarios and acknowledging the limitations of the ANLS hypothesis itself, would greatly enrich the discussion.
Author Response
The author demonstrates excessive bias throughout the text. He repeatedly disqualifies the scientific community that does not agree with his view on lactate. Instead of evaluating opposing evidence in a balanced manner, he attributes disagreements to mental habits and prejudices, which is an anti-scientific attitude.
The text frequently adopts a personal and emotional tone against other research groups. Terms such as "bias," "dogmatic," "twisting their brains," "scientific bias," and "unnecessary burden" appear repeatedly.
Although the manuscript correctly critiques the limitations of methods such as BOLD-fMRI, it does not deeply explore the legitimate reasons that have led scientists to question the ANLS hypothesis. References are selectively cited ("cherry-picking") to strengthen only the author's viewpoint.
The manuscript structure resembles more an opinion essay rather than a systematic review. There are no summary tables comparing supporting and opposing studies, nor a standardized critical analysis of the literature.
The author repeatedly cites his own previous works, which, although relevant, may convey a sense of self-affirmation bias.
Despite offering valuable contributions to the appreciation of lactate as an energy substrate, the article suffers from overconfidence in a single model (ANLS), underestimates methodological advances, and fails to recognize the metabolic plasticity of the brain. A more balanced approach, considering different physiological scenarios and acknowledging the limitations of the ANLS hypothesis itself, would greatly enrich the discussion.
I greatly appreciate the reviewer thorough review, critique and suggestions. I agree with the determination that my paper is not a review article, but rather an opinion paper. Thus, right at the Abstract section it is specifically indicated as such.
The “habit of mind” term has been coined by Margolis (ref. 17), and it does fit the situation where many persist to rely on the original, imperfect 1940 concept of glycolysis. This is despite the lack of scientific support for the claim that pyruvate is the end-product of aerobic glycolysis, while simultaneously do struggle to explain glycolytic lactate production in the presence of oxygen, naming it ‘aerobic glycolysis'. The strong stand that completely nullifies the role of lactate as an oxidative mitochondrial substrate, as if, somehow, there is competition between glucose and lactate, finds its roots in the NIH laboratory of the late Prof. Sokoloff’s, where glucose obligatory role as the energy substrate of the brain was established. I had the honor to know prof, Sokoloff, debate him and even have him admit to me that our original finding, showing the ability of lactate to solely support brain activity in vitro may prove to be a breakthrough.
The purpose of the paper is not to compare evidence for and against the ANLS hypothesis. Rather, it attempts to highlight the hurdles that prevent the acceptance of lactate for what it is. I argue that these hurdles are ‘habit of mind’ that afflict many in the field of brain energy metabolism, and a misplaced bias exhibited by the position of a leading scientific investigator who continually professes against any significant role for lactate in energy metabolism. That bias is exemplified in the list of papers (Table 1) published by the graduates of Sokoloff’s lab, most of them by G.A. Dienel. Gerald Dienel is a colleague and a friend with whom I participated in several conferences, where we had our debates on this topic more than once.
As to my own ‘bias,’ I am guilty as charged, especially where citing my own publications is concerned. However, I invite the reviewer to check Dienel’s own publications. For instance, in ref. 92, no less than 37 of the citations are of work by Dienel and his group. Ref. 90 includes over 20 citations of his work.
Where the ANLS hypothesis is concerned, for now, it is the only established hypothesis to offer a central role for lactate as a neuronal oxidative energy substrate. Using it in support of my argument about lactate role is just one of several others that are brought up in the paper’s text and are also included in the citations of my own work (bias?). Nonetheless, the work of Dienel et al., and that of others, which highlight the limitations of the ANLS hypothesis are also cited in the paper. In this respect, a legend was added to Table 1 that directly relates to this important point raised by the reviewer.
I hope that by declaring the paper to be an ‘opinion’, and by accepting responsibility for certain personal bias, as stated in the closing sentence of the abstract, the reviewer will agree to be more forgiving.
Round 2
Reviewer 2 Report
Comments and Suggestions for Authors
Thank you for your responses and for clarifying that your manuscript is intended as an opinion piece. However, I would like to point out that the journal classifies this submission as a Review Article, and it is labeled as such in the PDF provided for review.
This discrepancy creates confusion regarding the expectations for content structure, depth of analysis, and literature coverage. Additionally, several of the revisions I suggested in the previous round—particularly those aimed at strengthening the critical and integrative aspects of the text—were not addressed, under the justification that the manuscript reflects your personal viewpoint.
I believe it would be appropriate for the editorial team to assess whether the current format and intent of your manuscript align with the standards and scope of a Review Article in this journal.
Author Response
Thank you for your responses and for clarifying that your manuscript is intended as an opinion piece. However, I would like to point out that the journal classifies this submission as a Review Article, and it is labeled as such in the PDF provided for review.
This discrepancy creates confusion regarding the expectations for content structure, depth of analysis, and literature coverage. Additionally, several of the revisions I suggested in the previous round—particularly those aimed at strengthening the critical and integrative aspects of the text—were not addressed, under the justification that the manuscript reflects your personal viewpoint.
I believe it would be appropriate for the editorial team to assess whether the current format and intent of your manuscript align with the standards and scope of a Review Article in this journal.
My thanks to the reviewer for the thorough and helpful review. I emailed the Editor, asking to change the manuscript category from "Review" to "Opinion." I hope this will be done.
I agree with the reviewer's comment of the need to strengthen the integrative aspects of the text by adding the following text on page 12, while adding 10 more citations in support. I hope this will be satisfactory.
The ANLS hypothesis is not the only working concept that established, and later demonstrated through many studies, the use of lactate as the preferred neuronal oxidative substrate during brain activity. Only a sample of the increasing number of both human and animal studies showing brain utilization of lactate that originates from muscle and other tissues is cited here (102-111).
